# Transition path times of coupled folding and binding reveal the formation of an encounter complex

Flurin Sturzenegger[1], Franziska Zosel [1,4], Erik D. Holmstrom[1], Karin J. Buholzer[1], Dmitrii E. Makarov [2], Daniel Nettels[1] & Benjamin Schuler [1,3]

The association of biomolecules is the elementary event of communication in biology. Most mechanistic information of how the interactions between binding partners form or break is, however, hidden in the transition paths, the very short parts of the molecular trajectories from the encounter of the two molecules to the formation of a stable complex. Here we use single-molecule spectroscopy to measure the transition path times for the association of two intrinsically disordered proteins that form a folded dimer upon binding. The results reveal the formation of a metastable encounter complex that is electrostatically favored and transits to the final bound state within tens of microseconds. Such measurements thus open a new window into the microscopic events governing biomolecular interactions.

[1] Department of Biochemistry, University of Zurich, 8057 Zurich, Switzerland. [2] Department of Chemistry and Institute for Computational Engineering and Sciences, University of Texas at Austin, Austin, TX 78712, USA. [3] Department of Physics, University of Zurich, 8057 Zurich, Switzerland. [4]Present address: Novo Nordisk A/S, Novo Nordisk Park 1, 2760 Måløv, Denmark. Correspondence and requests for materials should be addressed to B.S. (email: schuler@bioc.uzh.ch)

nteractions between biomolecules are at the heart of all processes involving cellular signaling and communication. The mechanisms of how such interactions form are thus essential for both a fundamental understanding of these processes and targeted therapeutic intervention. Most of the mechanistic information is, however, contained in the exceedingly short parts of the reaction trajectories that start when the two molecules first encounter each other via translational diffusion and end with the formation of the stably bound complex. The transition states and intermediates visited on these transition paths are high in free energy and correspondingly unstable. Observing them experimentally has thus been challenging, and only in a few cases has it been possible to obtain glimpses of their structural or dynamic properties[1–7]. The experimental challenge is analogous to the one in protein folding: In both cases, the kinetics can often be approximated by a simple two-state reaction, where instantaneous transitions connect the initial and final states, but the most interesting information is hidden within the microscopic paths underlying these transitions[8,9]. Recent developments in single-molecule spectroscopy have started to reveal this information for transition paths in protein folding[10–13]. Here we show how these advances enable new ways of probing the transition paths of protein binding.

We investigate the association between the nuclear-coactivator binding domain (NCBD) of the CBP/p300 transcription factor and the activation domain of SRC-3 (ACTR), two members of the broad spectrum of intrinsically disordered proteins (IDPs), proteins that lack stable tertiary structure in isolation[14]. The interaction between ACTR and NCBD is a paradigm of coupled folding and binding[15,16], a mechanism that is frequently observed for IDPs. NCBD, a marginally stable, molten-globule-like IDP with pronounced helical content even in the unbound state[17], and the largely unstructured ACTR[15] bind to each other with nano-

molar affinity and form a cooperatively folded heterodimer[15]. We monitor their interaction using single-molecule Förster resonance energy transfer (FRET)[18] by labeling ACTR with a donor fluorophore, immobilizing it on a surface, and adding acceptor-labeled NCBD to the solution (Fig. 1a). Upon excitation, unbound ACTR emits only donor photons; on binding of an NCBD molecule, energy transfer results in a decrease in donor emission and an increase in acceptor emission (Fig. 1c). The signal change during the transition is recorded by confocal single-photon counting at high count rates (on average 200 ms$^{-1}$) to be able to probe microsecond timescales. This high time resolution allows us to measure binding transition path times and their distribution, which reveal the presence of an encounter complex with a lifetime of ~80 μs. The formation of this transient intermediate, where the molecules have associated but not yet folded, is favored by electrostatic interactions, in contrast to the final folding step. Such measurements thus create new opportunities for deciphering the mechanisms of protein binding.

## Results

**Measuring average transition path times.** We collected time traces of ACTR/NCBD binding events with the following approach (Fig. 1c, Supplementary Fig. 1 and 2): For each immobilized ACTR molecule, multiple association and dissociation events were first monitored at low excitation rate to ensure that the observed molecules are binding-competent, and to exclude a contribution from nonspecific surface interactions. Starting at a time when no NCBD was bound, the laser power was increased to the highest possible intensity that still allowed the observation of binding transitions with high photon rates before photobleaching occurred (Supplementary Fig. 1, see Methods). Dissociation events were not included since they exhibit the same

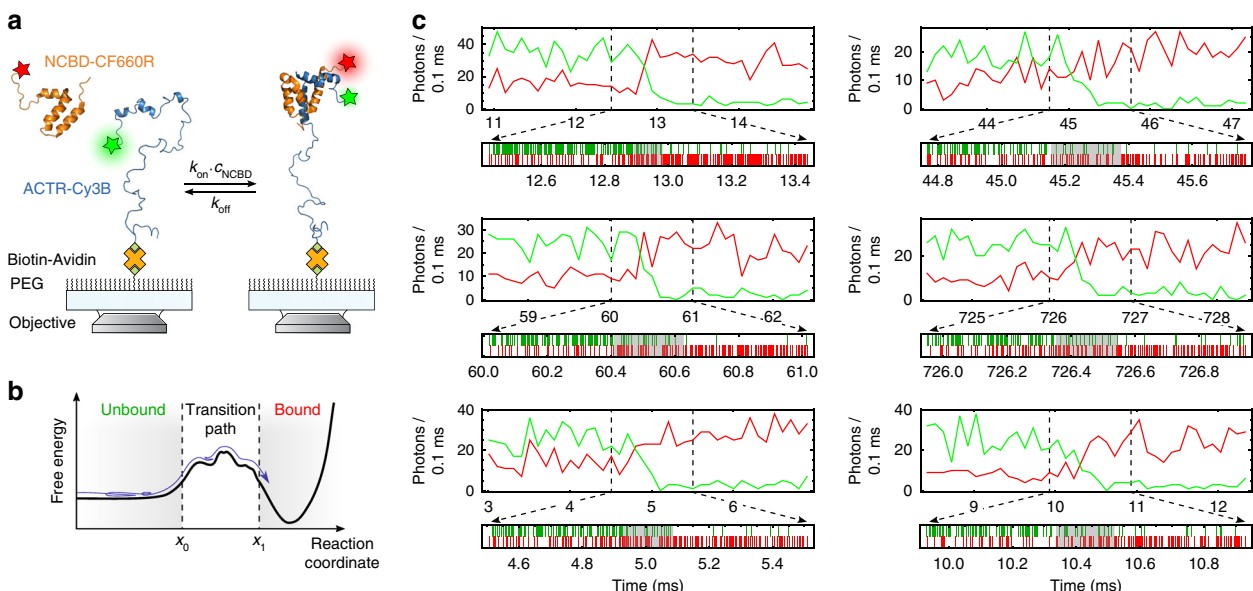

**Fig. 1** Observing coupled folding and binding by single-molecule FRET. **a** ACTR labeled with Cy3B as the donor fluorophore (green) is immobilized on a polyethyleneglycol (PEG)-coated quartz cover slide via a biotin—avidin—biotin linkage and excited by a laser beam. NCBD free in solution is labeled with CF660R as the acceptor fluorophore (red). Cartoon based on PDB entries 1KBH[15] and 2KKJ[17]. **b** Schematic free-energy landscape of the reaction with a molecular trajectory of a binding transition. The molecules are in the unbound or bound states most of the time and only rarely cross the energy barrier. A binding transition path extends from the time when $x_0$ is crossed until $x_1$ is reached without returning to $x_0$. **c** Examples of measured fluorescence time traces of binding events (ionic strength 108 mM, viscosity 1.28 cP), represented as binned (top) and single-photon data (bottom). The molecules start in the unbound state, indicated by high donor intensity (green) and low acceptor intensity (red). When NCBD binds to ACTR, the FRET efficiency increases in a rapid jump, corresponding to the transition path, with a concomitant increase in acceptor emission and a decrease in donor emission. In the single-photon representation of the transition-path region, donor photons are shown as green lines and acceptor photons as red lines. Segments of the trajectories identified as populating the transition path by the Viterbi algorithm are indicated by the gray shading in the single-photon time traces

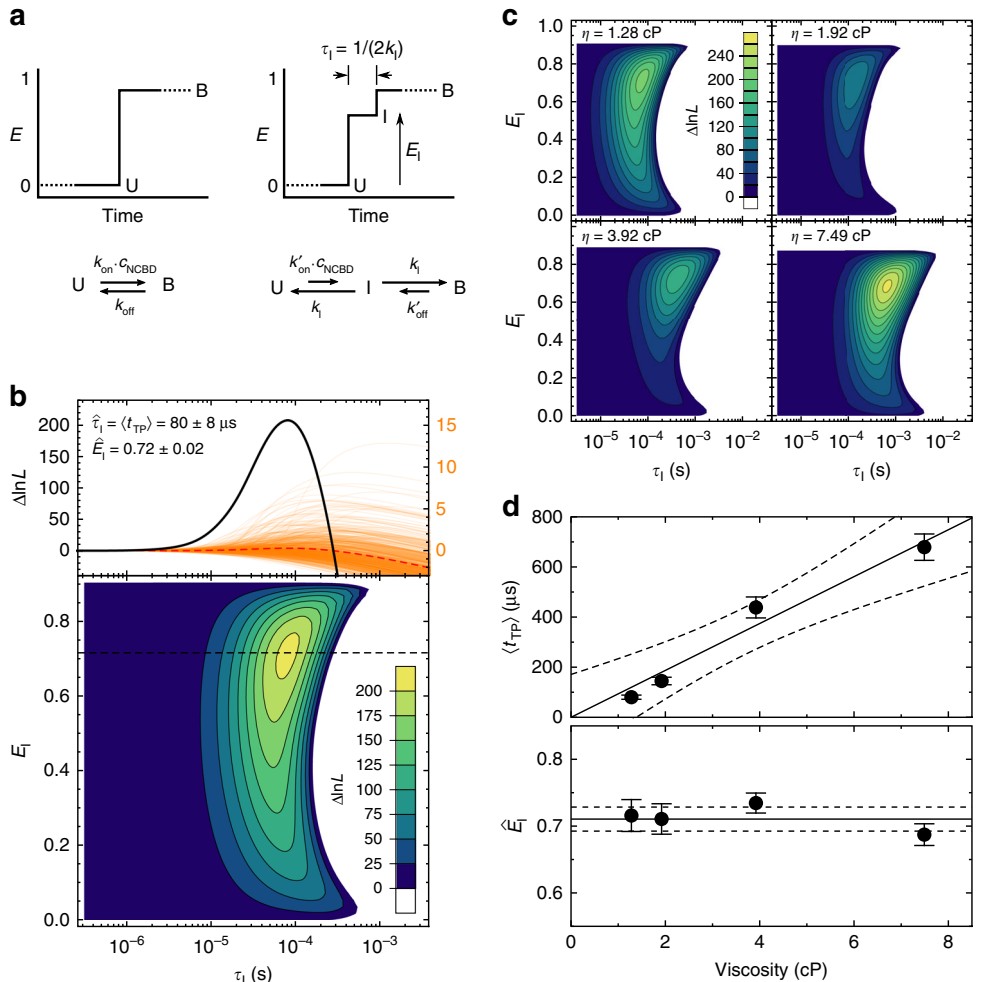

**Fig. 2** Quantifying transition path times. **a** The two kinetic models compared in the maximum likelihood analysis include the unbound (U) and bound (B) states; the second model accounts for a finite duration of a transition path by also including an intermediate state (I) with transfer efficiency $E_I$ and mean lifetime $\tau_I$. **b** Log likelihood difference ($\Delta \ln L$) plots for binding transitions (ionic strength 108 mM, viscosity 1.28 cP). The 2D contour plot (lower panel) shows the total $\Delta \ln L$ of all 686 transitions for different values of $\tau_I$ and $E_I$. The maximum yields $\hat{\tau}_I$ and $\hat{E}_I$, the most likely values of $\tau_I$ and $E_I$, respectively, with $\hat{\tau}_I = \langle t_{TP} \rangle$. The top graph shows the $\Delta \ln L$ plots for $\hat{E}_I$ (black dashed line in the contour plot). The $\Delta \ln L_j$ plots of all individual measurements are shown (orange lines, right scale), with their average (red dashed line, right scale) and their sum (black line, left scale). **c** $\Delta \ln L$ contour plots of transitions measured at different solvent viscosities ($\eta$). From the lowest to the highest viscosity, sample sizes were 686, 331, 285, and 378 transitions (see Supplementary Table 3). **d** Solvent viscosity dependence of $\langle t_{TP} \rangle$ (with linear fit, constrained to $\langle t_{TP} \rangle \geq 0$ at zero viscosity, and 90% confidence interval) and $\hat{E}_I$ (average: solid line; standard deviation: dashed lines). Error bars indicate standard errors obtained from 1000 bootstrapping trials

change in observed FRET efficiency as acceptor bleaching. Because of microscopic reversibility, however, the statistics of transition path times for dissociation should be identical to those for binding. Examples of recorded transitions are shown in Fig. 1c and in Supplementary Fig. 2. Despite the relatively high photon count rates, it is difficult to identify the start and end of a transition by visual inspection. For this reason, the photon arrival times were analyzed on a photon-by-photon level with the maximum-likelihood approach developed by Gopich and Szabo, which was previously applied to protein folding by Chung and Eaton[10,19,20].

In this analysis, the likelihood of two kinetic models is compared (Fig. 2a): A two-state model and a model with an intermediate state that mimics the transient species populated during the transition[10–12] (see Methods). By maximizing the log likelihood difference between the models, $\Delta \ln L$, with respect to the mean lifetime, $\tau_I$, and the transfer efficiency, $E_I$, of the intermediate state (Fig. 2b), we obtained the most likely values, $\hat{\tau}_I = 80 \pm 8\,\mu s$ and $\hat{E}_I = 0.72 \pm 0.02$, respectively, from the

analysis of all 686 measured transitions (unless stated otherwise ± indicates the standard error; here from 1000 bootstrapping trials). We tested the robustness of this result with two controls. To assess the influence of surface-attachment, we immobilized NCBD instead of ACTR. Using Cy3B-labeled NCBD on the surface and CF660R-labeled ACTR in solution (see Supplementary Fig. 3A), we obtained $\langle t_{TP} \rangle = 95 \pm 22\,\mu s$ and $\hat{E}_I = 0.66 \pm 0.05$, within experimental error of the above result. To assess the influence of the dyes, we measured immobilized Alexa488-labeled ACTR with CF660R-labeled NCBD in solution (see Supplementary Fig. 3B) and found $\langle t_{TP} \rangle = 101 \pm 11\,\mu s$, similar to the other dye pair; $\hat{E}_I = 0.52 \pm 0.03$ was lower than with Cy3B, as expected from the shorter Förster radius of Alexa488/CF660R (4.7 nm) compared to Cy3B/CF660R (6.0 nm).

The relatively high value of $\hat{E}_I$ indicates that the labeled C-terminal segments of ACTR and NCBD are in proximity already during the transition. The value of $\hat{\tau}_I$ can be interpreted as the average duration of the transition paths, $\langle t_{TP} \rangle$.[10–12] Remarkably, $\langle t_{TP} \rangle$ is very long compared to the transition path times of the

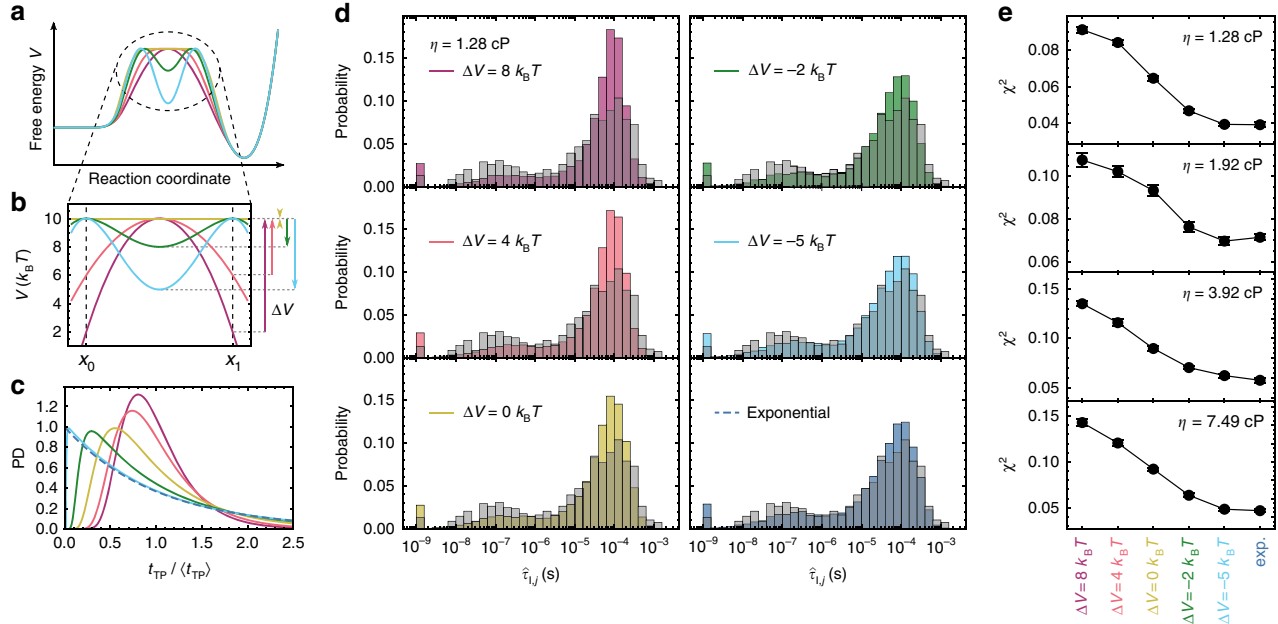

**Fig. 3** Obtaining the shape of the barrier from transition path time distributions. **a** Schematic of free-energy surfaces with different barrier shapes. **b** Potentials used to calculate transition path time distributions, with the relevant barrier heights or stabilities of the intermediate indicated by the arrows to the right (same color code used in all panels). **c** Probability density (PD) functions of transition path times, $t_{TP}$, calculated for the barrier shapes in (**b**), and an exponential $t_{TP}$ distribution (dashed). **d** Histogram of $\hat{\tau}_{I,j}$ for the measurement at 108 mM ionic strength without glycerol (gray), compared to histograms from simulated photon time traces based on different $t_{TP}$ distributions (colored, see legend). The smaller peaks in the sub-microsecond range are mostly due to transitions that were too fast to be accurately determined at the photon count rates of the corresponding time traces (see Methods and Supplementary Fig. 7). **e** $\chi^2$-distances between measured and simulated histograms for the measurements at different solvent viscosities, $\eta$ (standard errors from 27 simulations each)

folding of monomeric proteins previously observed in simulations[21] and most experiments[10–12]. The two most likely reasons for this surprisingly slow passage to the bound state are: (i) slow diffusion on the free-energy surface caused by internal friction arising from non-native interactions[12,13,22,23], or (ii) a local free-energy minimum corresponding to a high-energy intermediate or encounter complex[24–26], where ACTR and NCBD are already in contact but have not yet found their final, stably bound structure.

**Internal friction versus high-energy intermediate**. To examine the first possibility, we measured the dependence of $\langle t_{TP} \rangle$ on solvent viscosity, since a viscosity-independent component in the dynamics is a common fingerprint of internal friction[12,22,23]. The pronounced increase of $\langle t_{TP} \rangle$ with viscosity and the absence of a significant intercept at zero viscosity (Fig. 2c, d) suggest that internal friction does not substantially contribute to the transition-path dynamics. We note that $\hat{E}_I$—and thus the average inter-dye distance during the transition—exhibits no systematic change with viscosity, indicating that the addition of glycerol does not alter the conformational ensemble, attesting to the robustness of the analysis.

To probe the second possibility, the presence of a high-energy intermediate, as the cause for the long $\langle t_{TP} \rangle$, we take into account not only the average, but the distribution of transition path times, which is sensitive to the shape of the free-energy barrier[27,28]. The relatively large number of photons detected during the binding transitions enabled us to calculate $\Delta \ln L_j$ for every transition $j$ individually (orange curves in Fig. 2b) and identify the most likely transition path time, $\hat{\tau}_{I,j}$, for each. The resulting distribution of $\hat{\tau}_{I,j}$ (grey histograms in Fig. 3d) was then compared to the distributions expected for different barrier shapes. We tested barriers ranging from an inverted harmonic potential

(representing a simple transition state) to a flat barrier top and a high-energy intermediate with different stabilities (Fig. 3a, b) and calculated the corresponding $t_{TP}$ distributions for each potential by numerically solving the Smoluchowski equation[28] (Fig. 3c). The narrowest $t_{TP}$ distributions are those for a simple transition state, and they broaden with increasing stability of the intermediate. When the stability of the intermediate approaches a few $k_B T$, the transition path time becomes essentially equivalent to the lifetime of the intermediate, which, according to classical kinetics, should be exponentially distributed. Indeed, this is the behavior observed (Fig. 3c). To fully account for photon statistics, we simulated photon time traces with $t_{TP}$ sampled from these distributions, analyzed them in the same way as the measured data, and compared the resulting distributions of $\hat{\tau}_{I,j}$ to the experiment (Fig. 3d) by calculating the $\chi^2$-distance between them (Fig. 3e). The validity of this analysis was tested based on synthetic data (see Supplementary Fig. 4 and Methods for details).

At all solvent viscosities, the best agreement between simulation and measurement is achieved with a stability of the intermediate of at least $-5\,k_B T$ (Fig. 3d, e, Supplementary Fig. 5), where the $t_{TP}$ distribution is, within uncertainty, indistinguishable from an exponential distribution. This observation, together with the long $\langle t_{TP} \rangle$ and high $\hat{E}_I$ (Fig. 2), indicates the presence of an encounter complex where ACTR and NCBD are associated, but still separated by a free-energy barrier from the stably bound and folded state. Notably, the value of at least $5\,k_B T$ for the barrier height of escape from the encounter complex is consistent with an independent estimate based on Kramers theory[29]: From the reconfiguration time of ACTR, $\tau_r \approx 75$ ns, which has recently been measured[30], we estimate the preexponential factor to be $\tau_0 \approx 2\pi\tau_r \approx 0.5\,\mu s$.[31] Assuming two symmetrical barriers and using $\hat{\tau}_I = 80\,\mu s$ as the escape time, we obtain a barrier height of $\ln(2\hat{\tau}_I / \tau_0)\,k_B T \approx 5.8\,k_B T$. (We note that our measurements do not

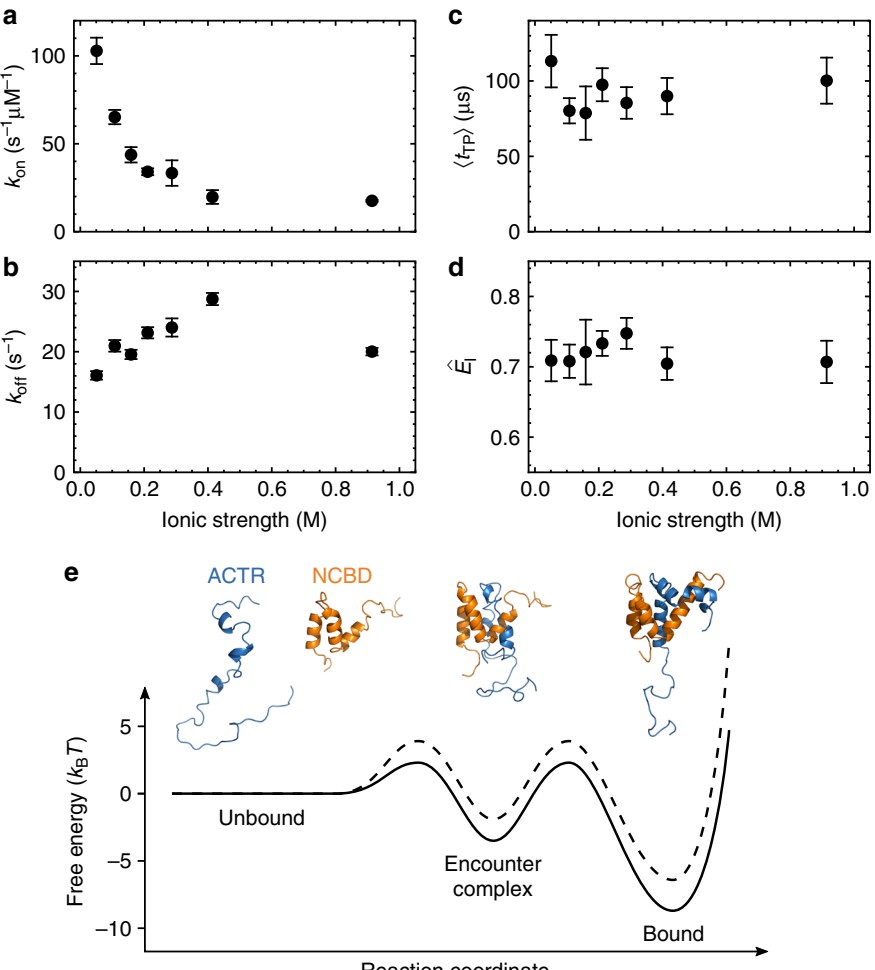

**Fig. 4** Ionic-strength dependence of ACTR-NCBD binding. **a** Association ($k_{on}$) and **b** dissociation rate coefficients ($k_{off}$) as a function of ionic strength from measurements at different NaCl concentrations. Error bars indicate standard errors obtained from 100 bootstrapping trials (sample sizes are listed in Supplementary Table 1). **c** Average transition path time, $\langle t_{TP} \rangle$, and **d** transfer efficiency in the encounter complex, $\hat{E}_I$, as a function of ionic strength. Error bars indicate standard errors from 1000 bootstrapping trials (sample sizes are listed in Supplementary Table 3). **e** Schematic of the coupled folding and binding process with free-energy surfaces in the absence (solid line) and presence (dashed line) of ~0.4 M salt. Cartoon based on PDB entries 1KBH[15] and 2KKJ[17]

allow us to determine the rate coefficients for the transitions from the intermediate to bound and the unbound states separately but only their sum.[10] The largest asymmetry of the barriers bounding the intermediate and still compatible with our results would be ~11 $k_B T$ versus ~5 $k_B T$ (see Maximum likelihood analysis of binding transitions in Methods).)

The shape of the observed $t_{TP}$ distributions can thus be explained by a localized intermediate. We note that roughness of the energy landscape along the reaction coordinate cannot account for our findings, although both scenarios would lead to a longer mean transition path time[12]. This is because introducing roughness, crudely speaking, amounts to lowering the effective diffusion coefficient along the reaction coordinate[32]. Indeed, numerical experiments where we introduced sinusoidal roughness of different amplitudes and calculated the resulting $t_{TP}$ distributions confirm that the roughness slows down the mean transition path time but has almost no effect on the shape of its distribution (see Supplementary Fig. 6).

**Ionic-strength dependence of transition path times.** To further investigate the nature of the encounter complex, in particular the role of electrostatics, we measured $\langle t_{TP} \rangle$, $\hat{E}_I$, and the association

and dissociation rate coefficients, $k_{on}$ and $k_{off}$, respectively, as a function of ionic strength. $k_{on}$ decreases about fivefold when the ionic strength is raised from 50 to 400 mM, while $k_{off}$ increases only about twofold (Fig. 4a, b), which is consistent with previous kinetic measurements[33] and in accord with the opposite net charge of ACTR and NCBD[34–36]. The encounter complex, however, behaves very differently: neither $\langle t_{TP} \rangle$ nor $\hat{E}_I$ change significantly with increasing ionic strength (Fig. 4c, d). The strong ionic-strength dependence of $k_{on}$ suggests that electrostatic interactions are formed already early during the transition, whereas the weak dependence of $\langle t_{TP} \rangle$ indicates that the subsequent formation of the folded state, which involves packing of the hydrophobic core, is less electrostatically driven.

## Discussion

In summary, the distribution of transition path times we have measured for the association of two IDPs reveal the presence of an encounter complex. Owing to this transient intermediate, it takes on average ~80 µs from the diffusional encounter of the two binding partners to the formation of the stably folded complex, much longer than the transition path time expected for the folding of a monomeric protein of similar size[10,11,21]. Neither

pronounced internal friction nor "roughness" of the free-energy surface, which have been shown to slow down some protein folding reactions involving non-native interactions or misfolding[12,13,22,23,37], are likely to be the cause of the long transition path times, as indicated by the strong solvent viscosity dependence of the transition path times we observe (Fig. 2d) and by simulations of the effect of energetic roughness on transition path time distributions (Supplementary Fig. 6).

The most likely mechanism for the coupled folding and binding of ACTR and NCBD is thus the initial formation of a transient encounter complex that is stabilized by the electrostatic interactions between the two oppositely charged IDPs[33], followed by folding (Fig. 4e). The barrier to encounter complex formation is very low, as reflected by an association rate coefficient that is only about an order of magnitude below the diffusion-limited value expected for a barrierless binding reaction[26,33,38,39]. Although the stability of the encounter complex slows the overall transition path time compared to simple monomeric folding reactions, it still proceeds remarkably quickly. The search process may be facilitated by relatively specific initial contacts between the N-terminal helices of ACTR and NCBD that provide non-covalent connectivity and reduce conformational freedom[36], as suggested by ϕ-value analysis[5], an increase in association rates with helicity[40], and simulations[41]. However, according to the ϕ-value analysis[5], these interactions are highly localized, and the extended hydrophobic interface between ACTR and NCBD is largely non-native near the transition state. This result implies that the electrostatic interactions favoring association[33] (Fig. 4a) are also predominantly non-native. However, if they interchange rapidly compared to the interconversion time to the native state and the net gain of electrostatic interactions upon folding is small, no pronounced dependence of $\langle t_{TP} \rangle$ on salt concentration is expected, in agreement with our observations (Fig. 4c).

In spite of the small size of ACTR and NCBD, their kinetics of folding and binding have been observed to be remarkably complex. Stopped-flow measurements revealed multistate kinetics on timescales of milliseconds to seconds[42], and recent single-molecule experiments identified a contribution due to peptidyl-prolyl cis/trans isomerization in the range of tens of seconds[39]. The timescale of 80 µs we observed here for the transition path time is similar to a fast kinetic phase observed during binding of ACTR to NCBD in temperature-jump experiments[43], which was attributed to the conformational exchange within NCBD previously identified by NMR[44]. The lifetime of the encounter complex might thus be linked to the internal dynamics of the molten-globule-like NCBD.

How do our observations for ACTR/NCBD relate to the behavior in other coupled folding-and-binding reactions of IDPs? The recent increase in kinetic investigations of IDP interactions has made it clear that the underlying mechanisms are diverse and difficult to generalize[35,45]. However, ACTR and NCBD do not seem to be an unusual case. With the high abundance of charged amino acids in IDPs[46], a pronounced role of electrostatics is commonly observed, especially for association rates[35,45]. An initial binding event that precedes folding also seems to be a common scenario[45], often referred to as "induced fit"[47]. However, even in cases where observations such as nonlinearities of concentration-dependent kinetics may indicate the presence of an encounter complex, characterizing its structural and dynamic properties has been more difficult, with NMR providing the most detailed insights so far[1,2]. Since the equilibrium populations of transient intermediates along the path of coupled folding and binding are typically low, detecting them with ensemble kinetics, such a temperature jump experiments[48], is challenging. Since much of the interesting mechanistic information is contained in the transition paths[13,20,49,50], probing them by single-molecule

spectroscopy provides an opportunity for revealing the mechanisms of protein binding and complementing kinetic and structural information from other methods. Next steps for advancing this approach will be to combine it with multiple labeling positions[38] or three-color FRET[51] to map the structure and dynamics of encounter complexes in protein interactions in more detail.

## Methods

**Protein expression**. *ACTR-Avi*: The coding sequence of a single-cysteine ACTR variant was cloned via *Bam*HI/*Hin*dIII into a pAT222-pD expression vector (gift from J. Schöppe and A. Plückthun)[52], yielding a protein construct with an N-terminal Avi-tag and a Thrombin-cleavable C-terminal His$_6$-tag (sequence of the cleaved construct: MAGLNDIFEA QKIEWHEGSM GSGSGTQNRP LLRNSLDDLV GPPSNLEGQS DERALLDQLH TLLSNTDATG LEEIDRALGI PELVNQGGQAL EPKQDCGGPR). pBirAcm (Avidity, Aurora CO, USA) was cotransfected for in vivo biotinylation of Lys12 in the Avi-tag[53], and expression was carried out in *Escherichia coli* C41(DE3) (Merck). Cells were grown at 37 °C in TYH medium (for 1 l: 20 g tryptone, 10 g yeast extract, 11 g HEPES, 5 g NaCl, 1 g MgSO$_4$, pH 7.3), supplied with 0.5% (w/v) glucose, until they reached an OD$_{600}$ of 0.8. Then, 50 µM biotin in 10 mM bicine buffer (pH 8.3) and 1 mM IPTG were added to the culture. Expression continued for 3 h at 37 °C, after which cells were harvested by centrifugation. The harvested cells were lysed by sonication, and the His$_6$-tagged protein was enriched via immobilized metal ion affinity chromatography (IMAC) on Ni-IDA resin (ABT). The His$_6$-tag was then cleaved off with thrombin (Serva Electrophoresis) and separated from the protein by another round of IMAC. Finally, biotinylated protein was separated from impurities and non-biotinylated protein via reversed-phase HPLC (RP-HPLC) on a C18 column (Reprosil Gold 200, Dr. Maisch, Germany) with a H$_2$O/0.1% trifluoroacetic acid −acetonitrile gradient. The purified protein was lyophilized, resuspended in buffer, and stored at −80 °C until use.

*ACTR*: ACTR containing a C-terminal cysteine was also inserted into the pAT222-pD expression vector containing an HRV 3C-cleavable N-terminal Avi-tag as well as a Thrombin-cleavable C-terminal His$_6$-tag (sequence of the cleaved construct: GPSGTQNRPL LRNSLDDLVG PPSNLEGQSD ERALLDQLHT LLSNTDATGL EEIDRALGIP ELVNQGGQALE PKQDCGGPR). ACTR was expressed like the other variants containing an N-terminal Avi-tag. After enrichment of the His$_6$-tag-containing protein by IMAC, the C-terminal His$_6$-tag was cleaved off, followed by a second round of IMAC. To obtain the fully cleaved protein, the Avi-tag was cleaved off by HRV-3C protease. The RP-HPLC purification was carried out as described above for two consecutive rounds.

*NCBD*: A construct with a single-cysteine residue and proline residues 20 and 23 replaced by alanine (to suppress kinetic heterogeneity due to peptidyl-prolyl cis/trans isomerization[39]) was generated by site-directed-mutagenesis (primers used (Microsynth): NCBD_P20A_P23A_fw: GCA TCT TCA GCG CAA CAG CAA CAG CAA GTT CTT AAC; NCBD_P20A_P23A_rev: GCT GTT GCG CTG AAG ATG CCG ATT TCA GCG TCC). Furthermore, the expression construct contained an N-terminal His$_6$-tag cleavable with HRV 3C protease (sequence of the cleaved construct: GPNRSISPSA LQDLLRTLKS ASSAQQQQQV LNILKSNPQL MAAFIKQRTA KYVANQPGMQ C). NCBD was coexpressed[54] with ACTR from a pET-47b(+) vector. Cell lysis and protein enrichment via IMAC were carried out as described above, followed by enzymatic cleavage of the His$_6$-tag with HRV 3C protease and separation of the tag from the proteins via another round of IMAC. Finally, ACTR and NCBD were separated with RP-HPLC as described above.

*NCBD-Avi*: The NCBD construct containing a single cysteine at the C-terminus as well as an N-terminal Avi-tag and a C-terminal cleavable His$_6$-tag was cloned, expressed and purified analogously to the ACTR-Avi variant (sequence of the cleaved construct: AGLNDIFEAQ KIEWHEGSMG SGSSPNRSIS PSALQDLLRT LKSASSAQQQ QQVLNILKSN PQLMAAFIKQ RTAKYVANQP GMQCGGPR). Also in this sequence, proline residues 20 and 23 were replaced by alanine to avoid kinetic heterogeneity.

**Protein labeling**. *ACTR-Avi*: Lyophilized protein was dissolved under nitrogen atmosphere to a concentration of 200 µM in 100 mM potassium phosphate buffer, pH 7.0, and was labeled for 3 h at room temperature with a 0.8-fold molar ratio of Cy3B or Alexa488 maleimide (GE Healthcare Life Sciences) to protein. Labeled protein was separated from unlabeled protein with RP-HPLC on a Sunfire C18 column (Waters) as described above.

*ACTR*: Lyophilized protein was dissolved to 50 µM in 50 mM sodium phosphate buffer, pH 7.0, and labeled with a 1.2-fold molar ratio of CF660R maleimide (Biotium) to protein. Labeled protein was separated from unlabeled protein with RP-HPLC on a Reprosil Gold 200 column, followed by RP-HPLC on a Sunfire C18 column.

*NCBD*: Lyophilized protein was dissolved to 170 µM in 100 mM potassium phosphate buffer, pH 7.0, and was labeled with a 1.2-fold molar ratio of CF660R to protein. Labeled protein was separated from unlabeled protein with RP-HPLC on a C18 column (Reprosil Gold 200), followed by RP-HPLC on a Sunfire column.

*NCBD-Avi*: Lyophilized protein was dissolved to 220 µM in 50 mM sodium phosphate buffer, pH 7.0, and labeled with a 1.2-fold molar ratio of Cy3B to

protein. Labeled protein was separated from unlabeled protein with RP-HPLC on a Reprosil Gold 200 column.

The correct mass of all labeled proteins was confirmed by electrospray ionization mass spectrometry.

**Sample preparation for surface experiments.** Surface experiments were performed using quartz cover slides coated with polyethylene glycol (PEG) and covalently modified with biotin (Quartz Coverslip (1″ × 1″), Bio 01, MicroSurfaces Inc., Englewood, NJ, USA). To clean the slides before use, they were boiled in water containing 0.1% Tween 20 and sonicated for 5 min. Silicone chambers (Secure Seal Hybridization Chambers, SKU:621202, Grace Bio Labs, Bend, OR, USA) were glued to the cover slide to yield four measurement chambers per slide. Biotinylated protein was immobilized on the cover slides with a biotin−avidin−biotin bond. To accomplish this, 200 µg/ml Avidin D (Vector Labs, Burlingame CA, USA) in NaP buffer (50 mM sodium phosphate, pH 7.0, 0.01% Tween 20) was added to the well and incubated for 3 min, followed by three washing steps with NaP buffer. Biotinylated protein was immobilized at a concentration of 10 pM in NaP buffer.

Measurements were performed in NaP buffer, with $H_2O$ replaced by $D_2O$ (NaP/$D_2O$) to increase the quantum yield of the dyes[55]. Compared to $H_2O$, the photon count rates in $D_2O$ were increased by ~20% for Cy3B and by ~50% for CF660R. To improve the signal quality further, we employed an oxygen scavenging system consisting of 400 U/ml bovine liver catalase (Sigma), 0.4 mg/ml glucose oxidase from *Aspergillus niger* (Sigma), and 1% (w/v) glucose, as well as a redox system (1 mM ascorbic acid, 1 mM methyl viologen)[56]. Concentrations of 20−80 nM acceptor-labeled NCBD or ACTR free in solution were used in the experiments.

To investigate the viscosity dependence, measurements were performed in NaP/$D_2O$ buffer containing 0, 14, 32, and 45% (v/v) glycerol. The viscosity was determined with a DV-I+ 4.0 Digital-Viscometer (Brookfield, Lorch, Germany). The ionic-strength dependence was measured in NaP/$D_2O$ buffer supplied with 0, 50, 100, 175, 300, and 800 mM NaCl. The point at 51 mM ionic strength was measured in 20 mM sodium phosphate, 10 mM NaCl, pH 7.0, 0.01% Tween 20 (in $D_2O$). The pH of each solution was set to 7.0 by adjusting the ratio of monobasic to dibasic phosphate.

**Instrumentation for surface experiments.** Surface experiments were performed on a MicroTime 200 confocal single-molecule instrument (PicoQuant, Berlin, Germany). A continuous-wave laser at 532 nm (LBX-532-50-COL-PP, Oxxius S.A., Lannion, France) was used for excitation. The light was focused into the sample (UplanApo 60/1.20W; Olympus, Japan), and the emitted light was collected with the same objective. A triple-band mirror (zt405/530/630rpc, Chroma, USA) and a long-pass filter (532 LP Edge Basic, Chroma) were used to separate the 532-nm laser light from the emitted fluorescence. The fluorescence light was then focused through a 100 µm pinhole and split by a dichroic mirror (T 635 LPXR, Chroma) to separate donor and acceptor photons. Donor emission was filtered with a 585/65 ET bandpass filter (Chroma), acceptor emission with a RazorEdge LP 647 RU long-pass filter (Chroma). Both photon streams were detected with avalanche photo-diode detectors (SPCM-AQR-15, PerkinElmer, Waltham MA, USA) and photon arrival times recorded with a HydraHarp 400 event timer (PicoQuant). The temporal resolution is limited by the random jitter of the detectors (~50 ps). A function generator (33600A Series Waveform Generator, Keysight Technologies, USA) connected to the modulation input of the laser driver allowed fast (<3 ms) and automated switching of the laser intensity (Supplementary Fig. 1). To scan the surface, the objective was mounted on a combination of two piezo-scanners, a P-733.2CL for XY-positioning and a PIFOC for Z-positioning (Physik Instrumente, Germany). To suppress oscillations of the scanner-stage, which can result in signal fluctuations, the digital notch filters were optimized for each axis.

**Analysis of long photon time traces.** To obtain the binding and unbinding rate coefficients, $k_{on}$ and $k_{off}$, as well as the transfer efficiencies of the unbound and bound states, $E_U$ and $E_B$, long photon time traces of surface-immobilized proteins were acquired at a laser power of 0.5 µW (measured at the back aperture of the objective). Time traces were inspected to ensure that no substantial brightness variations were occurring (e.g. caused by a drift of the molecule's position, long-lived dark states, or background fluctuations). Suitable traces were analyzed until photobleaching. Single-step photobleaching indicated that only one immobilized molecule was present in the confocal volume.

The pseudo-first-order association rate coefficient, $\bar{k}_{on} = k_{on} \cdot c_{NCBD}$, the dissociation rate coefficient, $k_{off}$, and the photon rates were determined using the maximum likelihood approach introduced by Gopich and Szabo[19]. $k_{on}$ is the second-order association rate coefficient, and $c_{NCBD}$ is the concentration of NCBD free in solution (for Supplementary Fig. 3A, where ACTR is free in solution, this would be $c_{ACTR}$). The likelihood of time trace $j$ is calculated from the general equation

$$L_j = \mathbf{p}_{fin}^T \prod_{i=1}^{N_j} \mathbf{n}_{c_i,j} \exp[(\mathbf{K} - \mathbf{n}_{D,j} - \mathbf{n}_{A,j})\tau_i] \mathbf{p}_{ini}, \tag{1}$$

where $N_j$ is the total number of photons in the time trace; $c_i$ is the color of the $i$th photon (D or A); $\tau_{i=1} = 0$, and $\tau_{i>1}$ is the inter-photon time, i.e. the time interval

between the detection of the $(i − 1)$th and $i$th photon. $\mathbf{K}$ is the rate matrix describing the association−dissociation dynamics. We include an additional dark state accounting for fluorophore blinking in the low-FRET unbound state, which is populated and depopulated with rate coefficients $k_{+b}$ and $k_{−b}$, respectively. Blinking also occurs in the high-FRET bound state but does not need to be included in the model, as it is not misrecognized as a transition. $\mathbf{K}$ is given by

$$\mathbf{K} = \begin{pmatrix} -(\bar{k}_{on} + k_{+b}) & k_{off} & k_{-b} \\ \bar{k}_{on} & -k_{off} & 0 \\ k_{+b} & 0 & -k_{-b} \end{pmatrix}. \tag{2}$$

$\bar{k}_{on}$ and $k_{off}$ are the rate coefficients of association and dissociation observed for immobilized ACTR at a given bulk concentration of NCBD. The dark state is populated and depopulated with rate coefficients $k_{+b}$ and $k_{−b}$, respectively. $\mathbf{n}_{D,j}$ and $\mathbf{n}_{A,j}$ in Eq. (1) are diagonal matrices with the observed donor photon rates ($n_{D,j}^U$, $n_{D,j}^B$, $n_{D,j}^{dark}$) and the acceptor photon rates ($n_{A,j}^U$, $n_{A,j}^B$, $n_{A,j}^{dark}$) of the three states on the diagonal, respectively. The photon rates vary slightly from time trace to time trace, mainly because the immobilized molecules are placed at slightly different positions inside the laser focus. $\mathbf{p}_{fin}^T = (1, 1, 1)$ is the transposed unity vector. The vector $\mathbf{p}_{ini}$ contains the populations at the start of the measurement. For the analysis of long time traces, we assume $\mathbf{p}_{ini} = \mathbf{p}_{eq}$, the equilibrium population of the three states, which is obtained from $\mathbf{K}\mathbf{p}_{eq} = 0$. We maximize $\sum_j \ln(L_j)$, the sum over the logarithms of the likelihoods of all photon time traces, with respect to $\bar{k}_{on}$, $k_{off}$, $k_{+b}$, $k_{−b}$, $\mathbf{n}_{D,j}$, and $\mathbf{n}_{A,j}$. For this purpose, we constrained the acceptor photon rate of the dark state to the acceptor photon rate of the unbound state, which is essentially the background signal of the acceptor detection channel ($n_{A,j}^{dark} = n_{A,j}^U$). Analogously, we constrained the donor photon rate of the dark state to the corresponding value of the bound state ($n_{D,j}^{dark} = n_{D,j}^B$), which is a good approximation since the transfer efficiency in the bound state is very high (i.e., $E_B \approx 0.9$).

To obtain the second-order association rate coefficient, $k_{on} = \bar{k}_{on}/c_{NCBD}$, the concentration of labeled protein free in solution needs to be known accurately. Because the concentrations can vary by up to 25% from experiment to experiment due to surface adhesion and pipetting errors, concentrations were determined directly in the sample with fluorescence correlation analysis[57]. The fluorescence of CF660R-labeled NCBD in solution was measured before and after each experiment, and the amplitude of the correlation curve was used to determine the average number of molecules inside the confocal volume, which is proportional to the concentration. The nominal concentrations were then corrected by the relative concentrations found from the correlation analysis.

For converting the photon count rates to transfer efficiencies, they need to be corrected for background fluorescence ($bg_A$ and $bg_D$), crosstalk between the detection channels (acceptor emission to donor channel, $\beta_{AD}$, and donor emission to acceptor channel, $\beta_{DA}$), acceptor direct excitation ($\alpha$), and differences in the quantum yields of the dyes and the detection efficiencies of the two channels ($\gamma$). We determined $bg_A$ and $bg_D$ for each time trace after the molecule had photobleached and corrected the measured photon count rates:

$$n'_{A,j} = n_{A,j} - bg_{A,j} \quad \text{and} \quad n'_{D,j} = n_{D,j} - bg_{D,j}. \tag{3}$$

$\beta_{AD}$ and $\alpha$ are negligible for these dye pairs in our instrument, so we can calculate the transfer efficiency as:

$$E_j = \frac{n'_{A,j} - \beta_{DA,j}n'_{D,j}}{n'_{A,j} - \beta_{DA,j}n'_{D,j} + \gamma n'_{D,j}}. \tag{4}$$

Since we know that $E_U = 0$, we can determine $\beta_{DA,j}$ and $\gamma_j$ directly from the measured time traces. In the unbound state, Eq. (4) simplifies to

$$n_{A,j}^{'U} - \beta_{DA,j}n_{D,j}^{'U} = 0, \tag{5}$$

and therefore

$$\beta_{DA,j} = \frac{n_{A,j}^{'U}}{n_{D,j}^{'U}}. \tag{6}$$

After correcting for $\gamma_j$, the total photon count rates in the bound and unbound state should be the same:

$$n_{A,j}^{'B} - \beta_{DA,j}n_{D,j}^{'B} + \gamma_j n_{D,j}^{'B} = n_{A,j}^{'U} - \beta_{DA,j}n_{D,j}^{'U} + \gamma_j n_{D,j}^{'U}. \tag{7}$$

Since $n_{A,j}^{'U} - \beta_{DA,j}n_{D,j}^{'U} = 0$, and we know $\beta_{DA,j}$, we can calculate $\gamma_j$:

$$\gamma_j = \frac{n_{A,j}^{'B} - \beta_{DA,j}n_{D,j}^{'B}}{n_{D,j}^{'U} - n_{D,j}^{'B}}. \tag{8}$$

We calculated $\beta_{DA,j}$ and $\gamma_j$ for each time trace and used them to calculate $E_{B,j}$ with Eq. (4). $E_{B,j}$ was then averaged over all time traces to get a mean transfer efficiency, $\langle E_B \rangle$. All parameters determined from long photon time traces are listed in Supplementary Tables 1 and 2.

**Measuring transition path times**. To measure transition path times, high-intensity time traces were recorded in an automated fashion. The piezo-driven scanning stage of the microscope allows surface-immobilized labeled proteins to be localized in a $20\,\mu m \times 20\,\mu m$ region of the cover slide. In the next step, the identified molecules are brought into focus one-by-one. The flow chart of the data acquisition procedure is detailed in Supplementary Fig. 1A. Initially, fluorescence is recorded at a laser power of $0.5\,\mu W$ (measured at the back aperture of the objective), and donor and acceptor photons are binned (binning interval 10 ms). If the photon count ratio $n_A / (n_A + n_D)$ is below 0.5 for five consecutive bins (i.e. no binding partner is bound), the laser is switched to high power ($5$–$50\,\mu W$) for 0.9 s in order to detect a potential binding event with much higher photon count rates. Afterwards, the laser power is switched off and the objective is moved to the next ACTR molecule. By always switching the laser to high power when the ACTR molecule is in the unbound state, we increase the probability of observing a binding transition instead of an unbinding transition during the period of high laser power. Additionally, the initial part of the recording at low laser power allows us to verify that the laser is indeed positioned on a functional molecule that shows anti-correlated changes in donor and acceptor signal characteristic of binding and unbinding. In Supplementary Fig. 1B, an example of a time trace with the switch between low and high laser power is shown.

We monitored only binding transitions because unbinding transitions exhibit the same change in observed FRET as acceptor photobleaching (in both cases the transfer efficiency drops to zero), which would bias the observed transition path times. In Supplementary Fig. 8, the log likelihood difference plots are compared for binding transitions, the unbinding/photobleaching transitions, and all transitions combined. For the unbinding/photobleaching transitions, no significant peak in the difference log likelihood is observed, as expected if transition paths for photobleaching are much faster than for unbinding.

**Analysis of high-intensity photon time traces**. The high-intensity time traces were inspected and transitions were identified visually. Around each transition, a time window was centered in a way that it did not contain any other transitions or blinking events. The duration of this window was chosen so that it was at least 1 ms long and contained at least 1000 photons. The range of the resulting window lengths is shown for each dataset in Supplementary Table 3. The photon count rates of the unbound and bound states were determined from the donor and acceptor emission before and after the transition.

To exclude time traces with blinking, blinking events were identified in the following way: For every detection channel and conformational state, the probability for each observed inter-photon time was calculated given the observed mean photon rate and number of photons, assuming exponentially distributed inter-photon times. Blinking events were defined as inter-photon times with a probability of less than 0.01, and the window used for analysis was chosen small enough to exclude all blinking events. If a blinking event occurred within less than 1 ms from the transition, the time trace was not used for analysis. The resulting photon time traces were then used for the maximum likelihood analysis. Supplementary Fig. 2 shows representative time traces, and Supplementary Table 3 shows for each dataset the number of analyzed transitions, the average total photon count rates, the range of window lengths, and the resulting $\langle t_{TP} \rangle$ and $\hat{E}_I$.

**Maximum likelihood analysis of binding transitions**. To obtain transition path times, we apply the method introduced by Chung and Eaton[10–12]. To approximate transition paths, a simple three-state model was used, where the transition path is described by a virtual intermediate state, I, between the unbound and bound states, U and B, respectively (here we assume that NCBD is in solution and ACTR immobilized; experiments with ACTR in solution and immobilized NCBD are described analogously):

$$U \underset{k_I}{\overset{k'_{on} \cdot c_{NCBD}}{\rightleftharpoons}} I \underset{k'_{off}}{\overset{k_I}{\rightleftharpoons}} B.$$

The depopulation of I is described by the rate coefficient $k_I$. The lifetime of the intermediate state, $\tau_I = 1/(2k_I)$, corresponds to the transition path time, $t_{TP}$. The rates from I to U and I to B are not necessarily equal; however, in our analysis we can only measure the lifetime of I, which is the inverse sum of the two rates. Since we only consider segments in the time traces where transitions from U to B occur, and we assume that we observe no $U \rightarrow I \rightarrow U$ and $B \rightarrow I \rightarrow B$ transitions, we set the rate coefficients for leaving I to be equal to simplify the analysis[10–12]. The rate coefficients to I from both directions are $k'_{on} \cdot c_{NCBD}$ and $k'_{off}$. They are related to $k_{on}$ and $k_{off}$ in a two-state model that assumes an instantaneous transition,

$$U \underset{k_{off}}{\overset{k_{on} \cdot c_{NCBD}}{\rightleftharpoons}} B,$$

via

$$k_{on} \approx \frac{1}{2} k'_{on} \quad \text{and} \quad k_{off} \approx \frac{1}{2} k'_{off}. \tag{9}$$

The factor 1/2 arises because the intermediate state can also react back to the original state in the three-state model, and so on average every second attempt leads to binding or unbinding.

The idea behind this approach of transition path time analysis is to compare the likelihood of an instantaneous transition, $L_j(\tau_I = 0)$, with the likelihood of an intermediate state with lifetime $\tau_I$ and transfer efficiency $E_I$, $L_j(\tau_I, E_I)$. Schematic FRET efficiency time traces for these two cases are shown in Fig. 2a. In both cases, the likelihood is calculated according to the general formula in Eq. (1). For $L_j(0)$, we use the rate matrix for the two-state model given by

$$\mathbf{K} = \begin{pmatrix} -\bar{k}_{on} & k_{off} \\ \bar{k}_{on} & -k_{off} \end{pmatrix}, \tag{10}$$

and for $L_j(\tau_I, E_I)$ we use the three-state model,

$$\mathbf{K} = \begin{pmatrix} -\bar{k}'_{on} & k_I & 0 \\ \bar{k}'_{on} & -2k_I & k'_{off} \\ 0 & k_I & -k'_{off} \end{pmatrix}, \tag{11}$$

where $\bar{k}_{on} = k_{on} \cdot c_{NCBD}$ and $\bar{k}'_{on} = k'_{on} \cdot c_{NCBD}$. To prevent random fluctuations in photon rate from being misrecognized as transitions to I, $\bar{k}_{on}$ and $k_{off}$ were set to $0.1\,s^{-1}$ and $\bar{k}'_{on}$ and $k'_{off}$ to $0.2\,s^{-1}$, which is slow compared to the average length of the fluorescence time traces. This approach is valid since we directly compare the two models.[10] Since we used only time traces starting in the unbound state and ending in the bound state, we have $\mathbf{p}_{ini}^T = (1, 0)$ or $\mathbf{p}_{ini}^T = (1, 0, 0)$, and $\mathbf{p}_{fin}^T = (0, 1)$ or $\mathbf{p}_{fin}^T = (0, 0, 1)$. For the two-state model, $\mathbf{n}_{D,j}$ and $\mathbf{n}_{A,j}$ are

$$\mathbf{n}_{D,j} = \begin{pmatrix} n_{D,j}^U & 0 \\ 0 & n_{D,j}^B \end{pmatrix} \quad \text{and} \quad \mathbf{n}_{A,j} = \begin{pmatrix} n_{A,j}^U & 0 \\ 0 & n_{A,j}^B \end{pmatrix}. \tag{12}$$

These rates are obtained from the photon rates in the individual time traces before and after the transition. For the three-state model, the corresponding matrices are

$$\mathbf{n}_{D,j} = \begin{pmatrix} n_{D,j}^U & 0 & 0 \\ 0 & n_{D,j}^I & 0 \\ 0 & 0 & n_{D,j}^B \end{pmatrix} \quad \text{and} \quad \mathbf{n}_{A,j} = \begin{pmatrix} n_{A,j}^U & 0 & 0 \\ 0 & n_{A,j}^I & 0 \\ 0 & 0 & n_{A,j}^B \end{pmatrix}. \tag{13}$$

The photon rates of the intermediate state of the three-state model are given by

$$n_{c,j}^I = n_{c,j}^U + \frac{E_I - \langle E_U \rangle}{\langle E_B \rangle - \langle E_U \rangle} \left( n_{c,j}^B - n_{c,j}^U \right) \quad \text{with} \quad c = A, D \tag{14}$$

where $\langle E_U \rangle$ is zero, and $\langle E_B \rangle$ was determined from long time traces acquired at low excitation power (see Analysis of long photon time traces).

The likelihoods for a time trace to have originated from an instantaneous transition or from a transition of finite duration, corresponding to an intermediate state I with lifetime $\tau_I$, and transfer efficiency $E_I$ can be calculated with Eq. (1). To compare the two models, the log likelihoods are subtracted,

$$\Delta \ln L_j(\tau_I, E_I) = \ln L_j(\tau_I, E_I) - \ln L_j(0), \tag{15}$$

and $\tau_I$ and $E_I$ are varied systematically to obtain log likelihood difference plots (Fig. 2b). $\Delta \ln L_j$ values of multiple time traces can be added to yield an average likelihood; from its maximum, the most likely lifetime, $\hat{\tau}_I = \langle t_{TP} \rangle$, and most likely transfer efficiency, $\hat{E}_I$, can be determined:

$$\Delta \ln L = \sum_j \Delta \ln L_j. \tag{16}$$

One can also find the most likely value for $t_{TP}$ of an individual transition by maximizing $\Delta \ln L_j$, although with higher uncertainty.

To test whether the peaks observed in the log likelihood difference plots originate from molecular binding rather than random fluctuations in the fluorescence signal, we used the following control: For each measured time trace, we deleted segments containing the transition and surrounding intervals of varying length (see Supplementary Fig. 9A). We analyzed these altered datasets again with the maximum likelihood method. If the likelihood peak is caused by the finite duration of the binding transition, then we expect the peak to disappear upon deleting the transition region. However, if it was due to fluctuations of the fluorescence signal, it would persist. In Supplementary Fig. 9B, the likelihood curves at $E_I = \hat{E}_I = 0.72$ are shown for time traces with segments of different lengths deleted. The likelihood peak disappears if more than about $250\,\mu s$ are deleted, indicating that the measured peak is caused by the transition path times of binding transitions. Upon deleting segments of $80\,\mu s$, corresponding to $\langle t_{TP} \rangle$, there is still a substantial peak, because many of the transitions are longer than $80\,\mu s$ (owing to the tail of the $t_{TP}$ distribution, Fig. 3).

To simplify the analysis, we assume the barriers for escape from I to be symmetric in height. However, we can estimate the maximum asymmetry in barrier heights from I to U and from I to B compatible with our experimental observations based on the following considerations (cf. Figure 4e): The ratio of about 0.1 between the observed association rate coefficient and a purely diffusion-

limited collision rate ($\sim 10^9\,\mathrm{M}^{-1}\,\mathrm{s}^{-1}$) yields an overall activation free-energy barrier for binding of $\sim 2.3\,k_BT$. From the observed dissociation rate and a preexponential factor of 0.5 μs (see main text), we obtain an overall activation free-energy barrier for dissociation of $\sim 11\,k_BT$ for the case of equal barriers from I to U and B (since in that case $k_{B\to I} = k'_{\rm off} = 2\,k_{\rm off} = 32\,\mathrm{s}^{-1}$). In this case, our estimate for the barrier heights for the escape from I is $\sim 5.8\,k_BT$ (see main text). These restraints correspond to the scenario shown in Fig. 4e (solid line). Our results would in principle, however, also be compatible with a situation where both the free energy of I and the barrier from I to B are reduced to the same extent. If we choose as a limit for this reduction the point where the free energies of I and B are equal, the barrier for I to U would be $\sim 11\,k_BT$. $k_{I\to U}$ would then be negligible compared to $k_{I\to B}$ for leaving I, in which case $1/k_{I\to B} = 80$ μs, resulting in a barrier from I to B of $\sim 5\,k_BT$. The largest asymmetry of the barriers bounding I would thus be $11\,k_BT$ versus $5\,k_BT$.

**Analysis of transition path time distributions.** To quantify the distribution of transition path times, we calculated $\Delta \ln L_j$ for every transition individually (using the $\hat{E}_I$ value found above), but for each transition $j$, we identified the $\hat{\tau}_{I,j}$ with the highest likelihood and generated a histogram from the resulting values (Fig. 3d). To find the underlying $t_{TP}$ distribution, we need to consider the broadening of the distribution due to the limited photon statistics. Photon time traces were thus simulated based on different theoretical transition path time distributions (see below) and analyzed in the same way as the measured data. To ensure that the photon statistics and shot-noise broadening are equivalent to those in the experimental data, we used the lengths and photon count rates of the measured time traces for the simulations. The simulations were performed in the following way: First, state trajectories were generated, each containing a single transition in the center, with a transition path time chosen randomly from the given theoretical distribution. For each state, photons were simulated with exponentially distributed inter-photon times, using the inverse of the total experimentally observed photon count rate of each state as mean inter-photon times. Photons were then randomly assigned to the acceptor or donor channel in accordance with the ratio of the corresponding photon count rates of the states observed experimentally. The photon count rates in the intermediate state were calculated using the $\hat{E}_I$ obtained from the measured data.

These simulated photon time traces were then analyzed in the same way as the measured time traces, and $\hat{\tau}_{I,j}$ was quantified for all transitions. The resulting $\hat{\tau}_{I,j}$ histogram, $H_s$, of the simulated data was then compared to the measured histogram, $H_m$, by calculating the $\chi^2$-distance:

$$\chi^2 = \sum_{i=1} \frac{(H_{m,i} - H_{s,i})^2}{H_{m,i} + H_{s,i}}. \qquad (17)$$

By simulating data with different theoretical $t_{TP}$ distributions and finding the one with the smallest $\chi^2$-distance to the measured data, we can identify the distribution of underlying transition path times that agrees best with the measured data. In addition to calculating the $\chi^2$-distance, we also performed a $k$-sample Anderson−Darling test[58] (see Supplementary Fig. 5), which tests whether two samples originate from the same underlying distribution, independent of the functional form of the distribution. Like the $\chi^2$-distance, this method finds the best agreement with our measured data for an exponential distribution. The accuracy of this analysis was tested based on simulations (see Brownian dynamics simulations of transition paths).

There are three peaks present in the $\hat{\tau}_{I,j}$ histograms (see Fig. 3d): The largest at $\langle t_{TP} \rangle$, a smaller one at $\sim 100$ ns, and a third one at $\sim 1$ ns. The one at 1 ns arises from all transitions lacking a maximum in their $\Delta \ln L_j$ plots, which are all collected in the shortest bin. The peak at $\sim 100$ ns also appears in the simulated datasets (see Fig. 3d), suggesting that it originates from the analysis of transitions that are too short or have too low a photon rate to be resolved accurately. To test this hypothesis, we simulated photon time traces of binding transitions with varying photon count rates and determined $\hat{\tau}_{I,j}$ histograms as for the experimental data (see Supplementary Fig. 7). Indeed, all three peaks are present in the simulated results, and the ones at $\sim 1$ ns and $\sim 100$ ns decrease in amplitude with increasing photon count rates. Even though this observation indicates that some transitions in our measurements are not resolved, the method of determining the barrier shape is still expected to be valid, since we take the limited photon statistics into account when simulating the photon time traces we compare to the experimental data (Fig. 3d).

**Theoretical transition path time distributions.** In a description of coupled folding and binding as Brownian motion on a 1D free-energy surface, the distribution of transition path times depends on the shape of the free-energy barrier and the effective diffusion coefficient. While the barrier shape determines the shape of the $t_{TP}$ distribution, the diffusion coefficient only determines the overall time-scale of the distribution. We tested different kinds of barrier shapes, including parabolic barriers of different heights, a flat barrier, and barriers with intermediates of different depths (Fig. 3a, b). We modeled the barriers with the following

equations:

$$\text{Barriers with transition state}: V(x) = -\frac{\Delta V}{x_1^2} x^2, \qquad (18)$$

$$\text{Flat barrier}: V(x) = 0, \qquad (19)$$

$$\text{Barriers with intermediate}: V(x) = -2\frac{\Delta V}{x_1^2} x^2 + \frac{\Delta V}{x_1^4} x^4. \qquad (20)$$

The transition path boundaries are $x_0$ and $x_1$, with $x_1 = -x_0$, and the heights or depths of the potentials are given by $\Delta V = V(0) - V(x_1)$.

From these functions, we calculated the $t_{TP}$ distributions numerically according to the procedure described in the appendix of ref. [28]. Specifically, this distribution is proportional to the flux of trajectories starting infinitely close to the left boundary, at $x = x_0 + \varepsilon$, and exiting through the right boundary, $x_1$, without returning to $x_0$:

$$p(t_{TP}) = \lim_{\varepsilon \to \infty} \frac{J(x_1, t_{TP})}{\int_0^\infty J(x_1, t)\mathrm{d}t}. \qquad (21)$$

The flux $J$ was computed by solving the Smoluchowski equation,

$$\frac{\partial p(x, t)}{\partial t} = -\frac{\partial J}{\partial x}, \qquad (22)$$

$$J(x, t) = -D\left(\beta V'(x) + \frac{\partial}{\partial x}\right)p(x, t), \qquad (23)$$

with absorbing boundary conditions, $p(x_0, t) = p(x_1, t) = 0$. The spectral expansion method was used to solve the Smoluchowski equation numerically: in this method, the Smoluchowski equation is first transformed into the equivalent Schrödinger equation, with absorbing boundaries being equivalent to introducing infinite potential walls at $x = x_0, x_1$. The Schrödinger equation is then solved by diagonalizing its effective Hamiltonian using particle-in-a-box wavefunctions as the basis.

For each $t_{TP}$ distribution calculated in this way, simulations of photon time traces were performed with different diffusion coefficients, corresponding to different values of $\langle t_{TP} \rangle$ (ranging from 90 to 130% of the measured $\langle t_{TP} \rangle$, in steps of 5%). Twenty-seven simulations were done for each value of $\langle t_{TP} \rangle$. $\hat{\tau}_I$ was determined for each simulation with the maximum-likelihood method, and for each set of 27, the average of all $\hat{\tau}_I$ was determined. The set with an average $\hat{\tau}_I$ closest to the experimentally observed $\langle t_{TP} \rangle$ was then used for comparison to the measured $\hat{\tau}_{I,j}$ histograms. The diffusion coefficients resulting in the best agreement with the measured values of $\langle t_{TP} \rangle$ are listed in Supplementary Table 4.

**Brownian dynamics simulations of transition paths.** To validate our method of finding the transition path time distribution, we performed Brownian dynamics simulations to generate transition paths for different barrier shapes. We then simulated fluorescence time traces based on these transition paths and analyzed them as described in Transition path time distribution analysis to test whether we could correctly identify the barrier shape on which the simulations were based. We used three different representative potentials for the Brownian dynamics simulations (see Supplementary Fig. 4A):

Barrier with transition state:

$$V(r) = 80\left((1.1(r-1))^4 - (1.1(r-1))^2\right). \qquad (24)$$

Flat barrier top:

$$V(r) = 80\left((1.1(r-1))^4 - (1.1(r-1))^2\right) - 6\mathrm{e}^{-2\left(\frac{20}{7}\right)^2 (r-1)^2}. \qquad (25)$$

Barrier with intermediate:

$$V(r) = 80\left((1.1(r-1))^4 - (1.1(r-1))^2\right) - \frac{44}{5}\mathrm{e}^{-2\cdot 5^2 (r-1)^2}. \qquad (26)$$

We adjusted the effective diffusion coefficient, $D$, for each potential so that $\langle t_{TP} \rangle$ between $r_0 = 0.8$ and $r_1 = 1.2$ (dashed lines in Supplementary Fig. 4A,B) was $\sim 80$ μs. We then simulated transitions with time steps of 0.1 μs (see Supplementary Fig. 4B) and converted distances into transfer efficiencies using the Förster equation:

$$E = \frac{1}{1 + (r/R_0)^6}, \qquad (27)$$

with the Förster radius $R_0 = 1$. The transfer efficiency time traces were then discretized into 20 states in steps of $\Delta E = 0.05$. To ensure that the photon statistics

are equivalent to those of the experimental data (dataset at 1.28 cP), we used the measured photon count rates to calculate the donor and acceptor photon count rates of each state,

$$n_{D,j}(E) = E\left(n_{D,j}^{B} - n_{D,j}^{U}\right) + n_{D,j}^{U} \quad \text{and} \quad n_{A,j}(E) = E\left(n_{A,j}^{B} - n_{A,j}^{U}\right) + n_{A,j}^{U}, \quad (28)$$

and simulated photon time traces based on the simulated state trajectories and the determined photon count rates as described in Transition path time distribution analysis. These photon time traces were then analyzed in the same way as the measured data. Both the $\chi^2$-distance and the $k$-sample Anderson−Darling test correctly identify the original barrier shapes (Supplementary Fig. 4C).

## Data availability

Data supporting the findings of this manuscript are available from the corresponding author upon reasonable request. A custom module for Mathematica (Wolfram Research) used for the analysis of single-molecule fluorescence data is available upon request.

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

## Acknowledgements

We thank Davide Mercadante, Attila Szabo, and Niels Zijlstra for discussions, Jendrik Schöppe and Andreas Plückthun for the pAT222-pD expression vector, Stefanie Joerg and Stephan Baumgartner for assistance with protein preparation, and the Functional Genomics Center Zurich for mass spectrometry analysis. This work was supported by the Swiss National Science Foundation (Grant Nos. 200021_169741, 310030B_173333, IZK0Z3_174658), the Robert A. Welch Foundation (Grant No. F-1514), and the US National Science Foundation (Grant No. CHE 1566001).

## Author contributions

Conceptualization, F.S., D.E.M., D.N., and B.S.; Methodology, F.S., F.Z., K.J.B., D.E.M., E.D.H., D.N., B.S.; Software, F.S., D.E.M., and D.N.; Investigation, F.S. and F.Z.; Formal analysis, F.S., D.E.M., and D.N.; Writing—Original Draft, F.S., D.N., and B.S.; Writing—Review and editing, F.S., F.Z., K.J.B., D.E.M., E.D.H., D.N., B.S.; Visualization, F.S., F.Z.; Funding acquisition, B.S. and D.E.M.; Resources, F.S., F.Z., E.D.H., K.J.B., and D.N.; Supervision, D.E.M., D.N., and B.S.

## Additional information

**Competing interests:** The authors declare no competing interests.

