## [Peer Review File · Nature Communications]

Reviewers' Comments:

Reviewer #1:

Remarks to the Author:

The manuscript presents a single molecule study of coupled folding-binding reaction between ACTR and NCBD. The study focuses on the binding mechanism and concludes for an intermediate state of an encounter complex with a relatively long life time. Using a comprehensive analysis and modeling the authors concluded that the slow transition path time is related to the high energy intermediate and not to internal friction arising from non-native interactions. From the dependence of the kinetics on ionic strength it is concluded that electrostatic forces govern the intermediate states.

Comments:

1. The conclusions regarding the encounter complex that is stabilized by electrostatic interactions but not by non-native interactions is somewhat contradictory. Very often, electrostatic interactions are nonspecific and are non-native in nature. If the electrostatic interactions were native, why is the life time of the encounter complex expected to be long? How can one validate the nature of the electrostatic interactions? Is it clear that some native salt bridges are formed in the encounter complex? Some mutations can shed light on the role of the electrostatic interactions.
2. The proposed mechanism for the binding-folding reaction is novel because two-state models have been used in the past to describe coupled folding-binding reactions. Two main differences between previous studies and the current one in the tethering of ACTR to a surface and the labeling of each protein. A central question is how changing the location of the fluorophores and the immobilization of one protein may affect the proposed mechanism.
3. Is the proposed mechanism expected to be relevant in vivo where the ionic strength is not minimal?
4. A more comprehensive survey of earlier studies of coupled folding binding reactions should be considered.

Reviewer #2:

Remarks to the Author:

Strutzenegger et al. discuss an exciting study of the encounter complex formed during coupled binding of folding of IDPs using single molecule FRET (smFRET). IDP binding is an important biological process with wide functional implications in cells. Induced binding via an encounter complex has been widely discussed, yet direct measurements of these short-lived species is difficult. The authors use smFRET to study this species for the binding of two biologically important proteins, NCBD domain of CBP and ACTR domain of SRC3. Using detailed analysis of smFRET traces, including with viscosity and salt titrations, they are able to directly test the characteristics of the encounter complex, and conclude that the species is likely separated by a barrier from the final bound state. The work is well done and systematically described with many appropriate controls. Given the importance and wide relevance of the problem and the interesting conclusions, I believe that the work will be of interest to the readers of Nature Communications. I only have a couple of minor suggestions listed below to further improve the manuscript.

Authors, please clarify what is the significance of the smaller peak around 100 ns in Figure 3D.

It would be useful to have a few examples of smFRET time-traces with high time-resolution in main text Figure 1, so that the formation of the complex can be directly visualized by the reader.

A brief note on the scanner used can be included in the instrumentation for surface experiments section.

Comment on the validity of using the same rate constant to both U and B from I in the 3-state MLE model (page 19).

Reviewer #3:

Remarks to the Author:

This paper is an excellent study of IDP folding/binding dynamics on the single molecule level that allows the experimenter to directly observe the transition between unbound and bound/folded states. The transition time, while too fast to directly observe in binned trajectories can be observed by maximum likelihood analysis of the photon arrival times, and finds a rate similar to the fastest protein folding times measured by bulk methods and faster than transition folding times measured by similar single-molecule methods. The transition time is independent of salt, which alters the association and dissociation times. Overall this is a significant result that will advance the field of IDP folding and binding, and I recommend publication after considering the following questions/suggestions:

1) The authors state that association/dissociation events were observed at low laser power to ensure the molecules are "fully functional." How do the authors determine functionality? I assume they are only checking that association happens, but can they determine correct fold or biological function?

2) Fig. 2B, top panel, the authors show curves for individual molecules and their sum, but it doesn't look to me like the orange curves would sum to the black curve, particularly 10^{-5} s region. Is the error very high in that region? The contour plot does a better job of showing the variation, perhaps the orange curves should be deleted.

3) Fig. 3 demonstrates that a sharp, monotonic peak in the energy landscape does not produce good agreement with the transition time distributions, but the best agreement is with an exponential probability density. But what does that mean in terms of the energy landscape and, more broadly, what does it tell you about the nature of the transition state?

4) I don't know why the authors separate "roughness" from the high energy intermediate, since all they measure is time, not equilibrium stability. These can be two sides of the same coin and the barrier could just be a series of small barriers rather than a dip in the barrier as shown in Fig. 3. There is no reason the roughness at the barrier need be the same as in the bound and unbound wells. The authors should explore these possibilities in the text.

Reviewer #4:

Remarks to the Author:

The manuscript by Sturzenegger describes an elegant set of experiments aimed at probing the association between an IDP (ACTR) and the MG-like protein NCBD. In particular, the authors use state of the art smFRET experiments to probe the rapid timescales of association from the unbound to the bound state, the so-called transition path time. A careful analysis of the experiments reveal that this process is best described as a two-step process, involving first forming an intermediate (an encounter complex), which then rearranges into the final complex. As such, the experiments provide central and until now unknown information about the timescales involved in binding of IDPs. Further, a detailed analysis of the distribution of transition path times provides insight into the shape of the barrier. The manuscript is well written, the experiments and

analyses carefully performed and described, and the conclusions appear well justified. As such, I only have few comments on the work that the authors might consider.

Major:

1. It would be nice to place the results a bit better in the context of what has already been measured for this system. The bulk kinetic experiments at low temperatures by Jemth has also been interpreted in terms of an on-pathway intermediate, but it is not clear whether this is the same and/or related. Also, in addition to the binding/unbinding kinetics, at least two other processes are known to occur in NCBD (major/minor state exchange and folding/unfolding), both occurring on the long-microsecond timescale. Presumably, by focusing only on direct binding events, these have no effect on the experiments here.
2. It is a bit unclear what the effect is of looking only at binding events. Obviously, at equilibrium the pathways should be fully reversible, and for a simple barrier crossing event the TP-time should also be the same. But in the current case, the authors interpret their data to represent what effectively is a high-energy intermediate (the encounter complex) with sizable barriers towards both unbound and bound states. In this case, it is not clear to me that looking only at binding reactions provides all information. Related, can the authors really make claims about the relative height of the two barriers going out of the intermediate (I->unbound vs I->bound)? These are assumed identical in the analysis, but I cannot see where this information would come from. Did the authors analyze any model with asymmetric barriers? Or is the symmetry merely a convenient assumption in the analysis of the data.
3. The authors' analysis shows that the intermediate has a stability of at least 5 kT. Is there any way of making an estimate of an upper bound? I guess that one could make some assumption of a lower bound of the pre-exponential factor (τ_0) so that e.g. $\tau_0 > 10\text{ns}$ would mean that the barrier is in between 5-10 kT. Presumably all of these would be compatible with the observed $p(\tau)$.
5. Martin Gruebele has interpreted a fast-timescale in T-jump of folding as depopulation of molecules sitting at the barrier. Can the authors estimate the equilibrium population in the intermediate and could it be populated at some concentration that could be probed?

Minor:

1. In addition to the papers cited on previous studies of encounter complexes (refs 1-5) the authors might consider citing the work of Marcellus Ubbink here (a review is cited later), and perhaps also the work from Nikolai Skrynnikov
2. As the authors discuss briefly it is unclear whether NCBD is best characterized as an IDP. The sentence "We investigate the association between the nuclear-coactivator binding domain (NCBD) of the CBP/p300 transcription factor and the activation domain of SRC-3 (ACTR), two members of the large group of intrinsically disordered proteins (IDPs), which lack stable tertiary structure in isolation¹²." might give the reader the impression that both ACTR and NCBD are fully unfolded prior to binding, but this is of course not the case. This could be particularly confusing given that the authors have recently studied a case where both binding partners were indeed fully unfolded.
3. The authors write "is recorded with microsecond time resolution". I would aid the reader a bit more, in particular those who are not experiments in this kind of high-powered smFRET and the resulting analyses. What is the actual time resolution?
4. The authors very nicely estimate relative concentrations by fluorescence correlation analysis in

the actual sample. It would be useful to know roughly how large these corrections are, to let the reader know what the effect would be of not doing this.

5. I find it a bit surprising in Figure S3C (with the simple barrier) that the p-value for $dV=8$ is lower than for $dV=4$ when the χ^2 is smaller.

Response to reviewers' comments for "Transition path times of coupled folding and binding reveal the formation of an encounter complex" by Sturzenegger *et al.*

We thank the reviewers for their thoughtful comments and their constructive criticism, which we have taken very seriously in our effort to improve the manuscript further. We have performed additional measurements, analysis, and simulations, clarified potentially ambiguous points, and extended the discussion. We hope that these changes are able to fully address the reviewers' concerns. Please find our detailed response below. The corresponding changes in the manuscript are marked in red.

Reviewers' comments:

Reviewer #1 (Remarks to the Author):

The manuscript presents a single molecule study of coupled folding-binding reaction between ACTR and NCBD. The study focuses on the binding mechanism and concludes for an intermediate state of an encounter complex with a relatively long life time. Using a comprehensive analysis and modeling the authors concluded that the slow transition path time is related to the high energy intermediate and not to internal friction arising from non-native interactions. From the dependence of the kinetics on ionic strength it is concluded that electrostatic forces govern the intermediate states.

Comments:

1. The conclusions regarding the encounter complex that is stabilized by electrostatic interactions but not by non-native interactions is somewhat contradictory. Very often, electrostatic interactions are nonspecific and are non-native in nature. If the electrostatic interactions were native, why is the life time of the encounter complex expected to be long? How can one validate the nature of the electrostatic interactions? Is it clear that some native salt bridges are formed in the encounter complex? Some mutations can shed light on the role of the electrostatic interactions.

We now realize that we may inadvertently have implied that the electrostatic interactions in the encounter complex are native interactions, presumably with our statement "The strong ionic-strength dependence of k_{on} suggests that electrostatic interactions stabilizing the complex are formed already early during the transition". In fact, as the reviewer suggests, the interactions are most likely transient and non-native. We changed the sentence to "The strong ionic-strength dependence of k_{on} suggests that electrostatic interactions are formed already early during the transition" and discuss our conclusions regarding the non-native character of electrostatics in the encounter complex more explicitly In the context of the ϕ values previously measured by Jemth and coworkers (p. 12):

"The search process may be facilitated by relatively specific initial contacts between the N-terminal helices of ACTR and NCBD that provide non-covalent connectivity and reduce conformational freedom³⁴, as suggested by ϕ -value analysis⁵, the increase in association rates with helicity³⁸, and simulations³⁹. However, according to the ϕ -value analysis⁵, these interactions are highly localized, and the extended hydrophobic interface between ACTR and NCBD is largely non-native at the transition state. This result implies that the electrostatic interactions favoring association³¹ (Fig. 4A) are also predominantly non-native. If they interchange rapidly compared to the interconversion time to the native state, and the net gain of electrostatic interactions upon folding is small, no

pronounced dependence of $\langle t_{TP} \rangle$ on salt concentration is expected, in agreement with our observations (Fig. 4C).”

2. The proposed mechanism for the binding-folding reaction is novel because two-state models have been used in the past to describe coupled folding-binding reactions. Two main differences between previous studies and the current one in the tethering of ACTR to a surface and the labeling of each protein. A central question is how changing the location of the fluorophores and the immobilization of one protein may affect the proposed mechanism.

To address this concern of robustness of our results we performed additional measurements. To test the influence of immobilization on transition path times, we inverted the experiment, i.e., immobilized NCBD instead of ACTR, and found very similar results, with $\langle t_{TP} \rangle = 95 \pm 22 \mu\text{s}$ and $\hat{E}_I = 0.66 \pm 0.05$. To further test the robustness with respect to the fluorophores used, we measured transition path times with immobilized ACTR labeled with a different donor dye, Alexa488 and observed $\langle t_{TP} \rangle = 101 \pm 11 \mu\text{s}$, close to the one measured with Cy3B ($80 \pm 8 \mu\text{s}$). Notably, the transfer efficiency of the intermediate, 0.51 ± 0.03 , is smaller for the Alexa488-CF660R pair compared to 0.72 ± 0.02 for Cy3B-CF660R, as expected given the shorter Förster radius for Alexa488-CF660R. Our results are thus quite independent of immobilization and the fluorophores used. We now included those measurements in the manuscript (p. 4/5 of the Results and Supplementary Fig. 3).

3. Is the proposed mechanism expected to be relevant *in vivo* where the ionic strength is not minimal?

Physiological ionic strengths are usually assumed to be in the range of 50-250 mM (e.g. Alberty, R. A., 2003. Thermodynamics of Biochemical Reactions. Wiley-Interscience, Hoboken, NJ.). The most extensive analysis we present is performed at an ionic strength of 108 mM (Figs. 2 & 3), and the ionic strength dependence covers 51 to 914 mM, so the proposed mechanism is expected to be relevant *in vivo*. We now mention the ionic strength explicitly in all figure captions to avoid ambiguity.

4. A more comprehensive survey of earlier studies of coupled folding binding reactions should be considered.

We agree that a more extensive survey of previous work on coupled folding and binding would be appropriate. We thank the reviewer for this suggestion and extended the Discussion section accordingly (p. 12/13).

Reviewer #2 (Remarks to the Author):

Struzzenegger et al. discuss an exciting study of the encounter complex formed during coupled binding of folding of IDPs using single molecule FRET (smFRET). IDP binding is an important biological process with wide functional implications in cells. Induced binding via an encounter complex has been widely discussed, yet direct measurements of these short-lived species is difficult. The authors use smFRET to study this species for the binding of two biologically important proteins, NCBD domain of CBP and ACTR domain of SRC3. Using detailed analysis of smFRET traces, including with viscosity and salt titrations, they are able to directly test the characteristics of the encounter complex, and

conclude that the species is likely separated by a barrier from the final bound state. The work is well done and systematically described with many appropriate controls. Given the importance and wide relevance of the problem and the interesting conclusions, I believe that the work will be of interest to the readers of Nature Communications. I only have a couple of minor suggestions listed below to further improve the manuscript.

Authors, please clarify what is the significance of the smaller peak around 100 ns in Figure 3D.

We thank the reviewer for the suggestion to address this point in more detail. The smaller peak around 100 ns is also present in the simulated data, suggesting that it is an intrinsic feature of the analysis. The peak decreases in amplitude if we simulate data with increasing photon count rates, indicating that it is due to transitions for which the exact transition path time could not be determined reliably with the photon count rates of the corresponding trajectories. We added a paragraph in the methods section "Analysis of transition path time distributions" (p. 28) and included the corresponding simulation results as Supplementary Figure 6 to illustrate this point.

It would be useful to have a few examples of smFRET time-traces with high time-resolution in main text Figure 1, so that the formation of the complex can be directly visualized by the reader.

We included more trajectories with high time resolution in Fig. 1. Even though the transition path times are much longer than the times previously observed for protein folding by Chung & Eaton, they are still too short for unequivocal observation by eye. A statistical analysis such as the maximum likelihood approach applied to a large data set is thus required for quantifying the transition path time and identifying evidence for an encounter complex. However, we now indicated with gray shading the segments identified by the Viterbi algorithm to be the most likely duration of the dwell time in the intermediate in the time traces in Fig. 1.

A brief note on the scanner used can be included in the instrumentation for surface experiments section.

A note on the scanner was included in the Methods section (p. 18).

Comment on the validity of using the same rate constant to both U and B from I in the 3-state MLE model (page 19).

Assuming the kinetic scheme

the lifetime of I , τ_I , is given by

$$\tau_I = \frac{1}{k_{-1} + k_2} .$$

Since in our experiments we can only measure τ_I , we cannot determine k_{-1} and k_2 independently. For the analysis of our data, we thus assume that $k_{-1} = k_2$ (analogous to the work of Chung & Eaton on protein folding). In principle, k_{-1} and k_2 could be quantified separately if we could observe transitions of the type $U \rightarrow I \rightarrow U$ or $B \rightarrow I \rightarrow B$. However, with the available time resolution, these excursions

cannot be distinguished from random fluctuations in the signal, and we have to rely on events where the transition occurs all the way from U to B. We added a short explanation in the chapter “Maximum likelihood analysis of binding transitions” in the Methods section (p. 23).

Our response to point 2 of Reviewer #4 addresses the closely related question of relative barrier heights for transitions from I to U and I to B. An estimate for the maximum asymmetry in barrier heights compatible with our experimental observations is given there.

Reviewer #3 (Remarks to the Author):

This paper is an excellent study of IDP folding/binding dynamics on the single molecule level that allows the experimenter to directly observe the transition between unbound and bound/folded states. The transition time, while too fast to directly observe in binned trajectories can be observed by maximum likelihood analysis of the photon arrival times, and finds a rate similar to the fastest protein folding times measured by bulk methods and faster than transition folding times measured by similar single-molecule methods. The transition time is independent of salt, which alters the association and dissociation times. Overall this is a significant result that will advance the field of IDP folding and binding, and I recommend publication after considering the following questions/suggestions:

1) The authors state that association/dissociation events were observed at low laser power to ensure the molecules are "fully functional." How do the authors determine functionality? I assume they are only checking that association happens, but can they determine correct fold or biological function?

Indeed – we only ensure that each immobilized molecule is able to bind NCBD and that the signal behaves as expected in terms of association/dissociation rates and signal stability. This is of course not enough to assure the correct fold or biological function in a broad sense, but it helps to eliminate molecules that are not functional in terms of being able to reversibly bind their partner. To clarify this point, we thus changed “fully functional” to “binding-competent” in the text (p. 4).

2) Fig. 2B, top panel, the authors show curves for individual molecules and their sum, but it doesn't look to me like the orange curves would sum to the black curve, particularly 10^{-5} s region. Is the error very high in that region? The contour plot does a better job of showing the variation, perhaps the orange curves should be deleted.

We agree that it may appear surprising that the orange curves sum to the black curve. Since there are 686 orange curves, there is of course a lot of overlap, and the visual impression is strongly influenced by the few curves with a maximum at long times. We tried different options to alter the representation and now chose a depiction where the individual orange curves are shown with reduced opacity, which makes it clearer that the curves with transition path times much greater than the mean are relatively few. We also added the average for all transitions as a red dashed line in Fig. 2B to stress the difference in scale for the orange curves and the black sum.

The contour plot in Fig. 2B shows the end result of the maximum likelihood analysis for the entire data set and is thus not suitable for illustrating the variability in the individual measurements.

3) Fig. 3 demonstrates that a sharp, monotonic peak in the energy landscape does not produce good agreement with the transition time distributions, but the best agreement is with an exponential

probability density. But what does that mean in terms of the energy landscape and, more broadly, what does it tell you about the nature of the transition state?

The exponential distribution is the limit for a stable intermediate, where we approach classical chemical kinetics. Fig. 3C shows that the t_{TP} -distribution of an intermediate with $\Delta V = -5kT$ is almost identical to an exponential distribution (with an additional sharp rise at very short times, because the transition path time cannot be zero). Indeed, in the χ^2 analysis, we cannot distinguish between the two distributions. Since any intermediate more stable than $-5kT$ would result in a distribution that is experimentally indistinguishable from an exponential, we can only set a lower bound on the stability from this analysis. We may not have made this point sufficiently clear in the text, so we have tried to explain this aspect more thoroughly (p. 7/8). Additionally, we have changed Fig. 3C, so that the exponential distribution is now shown as a dashed line to differentiate it from the distributions calculated for the free energy surfaces in Fig. 3A/B.

4) I don't know why the authors separate "roughness" from the high energy intermediate, since all they measure is time, not equilibrium stability. These can be two sides of the same coin and the barrier could just be a series of small barriers rather than a dip in the barrier as shown in Fig. 3. There is no reason the roughness at the barrier need be the same as in the bound and unbound wells. The authors should explore these possibilities in the text.

We thank the reviewer for bringing up this interesting point, which we now addressed in more detail and illustrate using simulations. It is of course correct that from the *average* transition path time, we cannot distinguish an intermediate from a rough barrier, since both will increase the time. However, from the *distribution* of t_{TP} , we can distinguish the two cases. The new Supplementary Fig. 7 illustrates this point: The introduction of roughness clearly shifts the distribution of t_{TP} (and thus $\langle t_{TP} \rangle$) to longer times, but in contrast to the case of an intermediate (Fig. 3C), the *shape* of the distribution remains essentially unchanged.

This observation is consistent with the view where energetic roughness renormalizes the diffusion coefficient (R. Zwanzig, Diffusion in a rough potential. *Proc. Natl. Acad. Sci.* 85, 2029–2030, 1988). In a way, this effect is part of our analysis, where the effective diffusion coefficient is an adjustable parameter. For instance, we need to assume a lower diffusion coefficient for the parabolic barrier compared to the case of an intermediate, to obtain the measured $\langle t_{TP} \rangle$. We now explicitly comment on this aspect in the main text (p. 9/10) and added Supplementary Fig. 7 and Supplementary Table 4 for clarification.

Reviewer #4 (Remarks to the Author):

The manuscript by Sturzenegger describes an elegant set of experiments aimed at probing the association between an IDP (ACTR) and the MG-like protein NCBD. In particular, the authors use state of the art smFRET experiments to probe the rapid timescales of association from the unbound to the bound state, the so-called transition path time. A careful analysis of the experiments reveal that this process is best described as a two-step process, involving first forming an intermediate (an encounter complex), which then rearranges into the final complex. As such, the experiments provide central and until now unknown information about the timescales involved in binding of IDPs. Further, a detailed analysis of the distribution of transition path times provides insight into the shape of the barrier. The manuscript is well written, the experiments and analyses carefully performed and described, and the conclusions appear well justified. As such, I only have few comments on the work that the authors might consider.

Major:

1. It would be nice to place the results a bit better in the context of what has already been measured for this system. The bulk kinetic experiments at low temperatures by Jemth has also been interpreted in terms of an on-pathway intermediate, but it is not clear whether this is the same and/or related. Also, in addition to the binding/unbinding kinetics, at least two other processes are known to occur in NCBD (major/minor state exchange and folding/unfolding), both occurring on the long-microsecond timescale. Presumably, by focusing only on direct binding events, these have no effect on the experiments here.

The reviewer raises an interesting point, as fast kinetic phases have indeed been observed in the binding between ACTR and NCBD. Most of these experiments have been conducted at 4°C, but an extrapolation to room temperature yields rates around 5000-10000 s⁻¹, close to the timescale of 100 μs we observe here. This timescale also coincides with the timescale of conformational exchange within NCBD, so it is not unreasonable to suspect that this timescale also limits the speed with which the encounter complex decays. Accordingly, we included a discussion of this point in the main text (p. 12/13).

2. It is a bit unclear what the effect is of looking only at binding events. Obviously, at equilibrium the pathways should be fully reversible, and for a simple barrier crossing event the TP-time should also be the same. But in the current case, the authors interpret their data to represent what effectively is a high-energy intermediate (the encounter complex) with sizable barriers towards both unbound and bound states. In this case, it is not clear to me that looking only at binding reactions provides all information. Related, can the authors really make claims about the relative height of the two barriers going out of the intermediate (I→unbound vs I→bound)? These are assumed identical in the analysis, but I cannot see where this information would come from. Did the authors analyze any model with asymmetric barriers? Or is the symmetry merely a convenient assumption in the analysis of the data.

As the reviewer points out, the TP-time should be the same for both directions, no matter whether there is a single barrier, an intermediate, or any other barrier shape, because of time reversal symmetry. In the case of an intermediate state, the TP-time can be assumed to be equal to the lifetime of the intermediate state, and this lifetime is the same, independent of whether the molecules go to the unbound or to the bound state. As pointed out in response to Reviewer 2, we thus cannot determine the two rates independently and assume equal rate coefficients or barrier heights for I to U and I to B as the simplest case.

However, we can estimate the maximum asymmetry in barrier heights from I to U and from I to B compatible with our experimental observations based on the following considerations (cf. Fig. 4E): The ratio of about 0.1 between the observed association rate coefficient and a purely diffusion-limited collision rate (~10⁹ M⁻¹s⁻¹) yields an overall activation free energy barrier for binding of ~2.3 k_BT. From the observed dissociation rate and a preexponential factor of 0.5 μs (see main text), we obtain an overall activation free energy barrier for dissociation of ~11 k_BT for the case of equal barriers from I to U and B (since in that case $k_{B \rightarrow I} = k'_{off} = 2 k_{off} = 32 \text{ s}^{-1}$). In this case, our estimate for the barrier heights for the escape from I is ~5.8 k_BT (see main text). These restraints correspond to the scenario shown in Fig. 4E (solid line). Our results would in principle, however, also be compatible with a case where both the free energy of I and the barrier from I to B are reduced to the same extent. If we choose as a limit for this reduction the point where the free energies of I and B are equal, the barrier for I to U would be ~11 k_BT. $k_{I \rightarrow U}$ would then be negligible compared to $k_{I \rightarrow B}$ for

leaving *I*, in which case $1/k_{I \rightarrow B} = 80 \mu\text{s}$, resulting in a barrier from *I* to *B* of $\sim 5 k_B T$. The largest asymmetry of the barriers bounding *I* would thus be $11 k_B T$ versus $5 k_B T$.

We now mention this aspect regarding asymmetric barriers explicitly in the Results section (p. 9) and include these considerations in detail in the Methods section (p. 26/27).

3. The authors' analysis shows that the intermediate has a stability of at least 5 kT. Is there any way of making an estimate of an upper bound? I guess that one could make some assumption of a lower bound of the pre-exponential factor (τ_0) so that e.g. $\tau_0 > 10\text{ns}$ would mean that the barrier is in between 5-10 kT. Presumably all of these would be compatible with the observed $p(\text{tp})$.

This is a very good suggestion that certainly deserves more discussion. Since for any intermediate more stable than $\sim 5 k_B T$, the transition path time distribution looks exponential, we can indeed only give a lower bound based on the TP-time distribution analysis alone. Triggered by this question, we now decided to refine our discussion of this point:

Instead of just assuming a generic pre-exponential factor of $1 \mu\text{s}$, we more specifically use the reconfiguration time of ACTR, which was recently measured to be $\tau_r \approx 75 \text{ns}$ (A. Soranno, F. Zosel, H. Hofmann, *J. Chem. Phys.* 148, 123326, 2018). Based on Kramers theory, the preexponential factor can be estimated from this quantity as $\tau_0 = 2\pi \tau_r \approx 0.5 \mu\text{s}$ (Nettels et al., *Proc. Natl. Acad. Sci. USA* 104, 2655-2660, 2007), resulting in a barrier height of $\sim 5.8 k_B T$.

In view of the discussion in response to the Reviewer's point 2, we think that yet another estimate of barrier height based on assuming a different (and more arbitrary) prefactor may be confusing, but given the explicit treatment we now included, it will be easy for the reader to make other estimates. If, e.g., we assume 10 ns for the prefactor, the resulting barrier can indeed not be greater than $\sim 10 k_B T$, as suggested by the reviewer.

5. Martin Gruebele has interpreted a fast-timescale in T-jump of folding as depopulation of molecules sitting at the barrier. Can the authors estimate the equilibrium population in the intermediate and could it be populated at some concentration that could be probed?

This is another interesting point. Assuming the kinetic scheme

and using our results, $k'_{on} \approx 2k_{on} = 206 \text{s}^{-1} \mu\text{M}^{-1}$, $k'_{off} \approx 2k_{off} = 32 \text{s}^{-1}$ and $k_I = 1/(2\tau_I) = 6250 \text{s}^{-1}$ we can estimate the equilibrium population. If we assume a high protein concentration, such that the molecules are almost always in the bound state (above $\sim 10 \mu\text{M}$), so that *I* is maximally populated, the fraction of *I* is $\sim 0.5\%$, probably too low to detect in T-jump experiments. We now included this aspect in the Discussion section (p. 13).

Minor:

1. In addition to the papers cited on previous studies of encounter complexes (refs 1-5) the authors might consider citing the work of Marcellus Ubbink here (a review is cited later), and perhaps also the work from Nikolai Skrynnikov.

We thank the reviewer for these suggestions and have included references to their work in the introduction (p. 2).

2. As the authors discuss briefly it is unclear whether NCBD is best characterized as an IDP. The sentence “We investigate the association between the nuclear-coactivator binding domain (NCBD) of the CBP/p300 transcription factor and the activation domain of SRC-3 (ACTR), two members of the large group of intrinsically disordered proteins (IDPs), which lack stable tertiary structure in isolation¹².” might give the reader the impression that both ACTR and NCBD are fully unfolded prior to binding, but this is of course not the case. This could be particularly confusing given that the authors have recently studied a case where both binding partners were indeed fully unfolded.

We agree that being more explicit about the location of ACTR and NCBD within the broad spectrum of IDPs will be useful and rephrased our statement accordingly: “...two members of the broad spectrum of intrinsically disordered proteins (IDPs), proteins that lack stable tertiary structure in isolation¹⁴. Their interaction is a paradigm of coupled folding and binding^{15,16}, a mechanism that is frequently observed for IDPs. NCBD, a marginally stable, molten-globule-like IDP with pronounced helical content even in the unbound state¹⁷, and the largely unstructured ACTR¹⁵ bind to each other with nanomolar affinity and form a cooperatively folded heterodimer¹⁵.”

3. The authors write “is recorded with microsecond time resolution”. I would aid the reader a bit more, in particular those who are not experiments in this kind of high-powered smFRET and the resulting analyses. What is the actual time resolution?

Answering this question unequivocally is less straightforward than one might think. The instrument records photon arrival times with an accuracy of ~50 ps (limited by the random jitter of the detectors), so in some sense this is the time resolution. However, to be able to make any inference about changes in transfer efficiency (or distance), we need information from multiple photons. Here we thus wanted to stress the high photon count rates (which on average are about 200 ms⁻¹) required to probe timescales in the microsecond regime. To be more explicit, we thus now changed “is recorded with microsecond time resolution” to “is recorded by confocal single-photon counting at high count rates (on average 200 ms⁻¹) to be able to probe microsecond timescales” (p. 3) and separately specify the 50 ps photon timing accuracy of the instrument in the Methods section (p. 18).

4. The authors very nicely estimate relative concentrations by fluorescence correlation analysis in the actual sample. It would be useful to know roughly how large these corrections are, to let the reader know what the effect would be of not doing this.

Especially when using low protein concentrations, the variation from experiment to experiment can be up to ~25%. We now added this information in the methods section.

5. I find it a bit surprising in Figure S3C (with the simple barrier) that the p-value for $dV=8$ is lower than for $dV=4$ when the χ^2 is smaller.

Since the two methods of comparing the histograms are not identical, they can have slightly different results. The overall trends are, however, the same for both methods, and the difference is close to the estimated uncertainty in the p-value.

Reviewers' Comments:

Reviewer #1:

Remarks to the Author:

All my questions have been satisfactorily addressed.

Reviewer #2:

Remarks to the Author:

The authors have done a very good job of revising the manuscript to address my earlier review comments. This work provides novel insight into the biophysics of IDPs and will be of high interest to the readers of Nature Communications.

Reviewer #3:

Remarks to the Author:

The authors have addressed all my questions and I recommend publication.

Reviewer #4:

Remarks to the Author:

The authors have done an excellent job in responding to my questions/comments, and as far as I can see also to those of the other reviewers. The paper was already very nice, and still is. I have no further comments.